# Enhancing Behavioural Changes: A Narrative Review on the Effectiveness of a Multifactorial APP-Based Intervention Integrating Physical Activity

**DOI:** 10.3390/ijerph21020233

**Published:** 2024-02-16

**Authors:** Giulia Di Martino, Carlo della Valle, Marco Centorbi, Andrea Buonsenso, Giovanni Fiorilli, Giuseppe Calcagno, Enzo Iuliano, Alessandra di Cagno

**Affiliations:** 1Department of Medicine and Health Sciences, University of Molise, 86100 Campobasso, Italy; g.dimartino1@studenti.unimol.it (G.D.M.); carlo.dellavalle@univr.it (C.d.V.); m.centorbi@studenti.unimol.it (M.C.); andrea.buonsenso@unimol.it (A.B.); fiorilli@unimol.it (G.F.); 2Department of Neurosciences, Biomedicine and Movement, University of Verona, 37129 Verona, Italy; 3Faculty of Medicine, University of Ostrava, 70103 Ostrava, Czech Republic; enzo.iuliano@uniecampus.it; 4Faculty of Psychology, eCampus University, 22060 Novedrate, Italy; 5Department of Movement, Human and Health Sciences, University of Rome “Foro Italico”, 00135 Rome, Italy; alessandra.dicagno@uniroma4.it

**Keywords:** chronic disease, telemedicine, eHealth, weight control, physical exercise

## Abstract

The rapid evolution of technologies is a key innovation in the organisation and management of physical activities (PA) and sports. The increase in benefits and opportunities related to the adoption of technologies for both the promotion of a healthy lifestyle and the management of chronic diseases is evident. In the field of telehealth, these devices provide personalised recommendations, workout monitoring and injury prevention. The study aimed to provide an overview of the landscape of technology application to PA organised to promote active lifestyles and improve chronic disease management. This review identified specific areas of focus for the selection of articles: the utilisation of mobile APPs and technological devices for enhancing weight loss, improving cardiovascular health, managing diabetes and cancer and preventing osteoporosis and cognitive decline. A multifactorial intervention delivered via mobile APPs, which integrates PA while managing diet or promoting social interaction, is unquestionably more effective than a singular intervention. The main finding related to promoting PA and a healthy lifestyle through app usage is associated with “behaviour change techniques”. Even when individuals stop using the APP, they often maintain the structured or suggested lifestyle habits initially provided by the APP. Various concerns regarding the excessive use of APPs need to be addressed.

## 1. Introduction

Telehealth is focused on fostering the long-term wellness of patients and facilitating the self-management of health. In accordance with the American Telemedicine Association, this technological solution empowers care providers to offer remote family support by enabling them to monitor, educate, gather data and provide innovative multidisciplinary care, especially in the organisation and management of physical activities (PAs) [1]. An active lifestyle not only enhances well-being and promotes good health through non-pharmacological interventions but also prevents the long-term development of diseases such as cardiovascular problems, diabetes, osteoporosis and obesity [2]. To achieve health benefits, exercise may be regularly performed for an extended period, which is also significantly associated with positive mood and life satisfaction [3]. The dropout rate, a general issue in numerous longitudinal trials involving PA interventions, tends to be lower when utilizing tele-exercise, as underscored by a recent study [4]. The most common difficulties in regularly engaging in PA are the limited access to sports facilities, the challenge of reaching them, a lack of time and the costs associated with fitness programmes [5]. The spread of smart devices, including wearables and environmental sensors, has made it easier to gather real-time data on PA users as well as vital physiological metrics [6]. In the field of telehealth, the significance of these devices has been highlighted for health and wellbeing purposes, including workout data, to offer personalised suggestions, monitor workouts and prevent injuries.

The support for technological devices ranges from the simplest ones, like pedometers analysing daily step counts, through to those where exercise protocols are suggested and pre-recorded in some smartphone applications, up to live-streamed exercise programmes [7,8,9,10]. Pedometers provide each person with feedback on their daily activity (walking, sitting, standing, gardening and pet walking) [11]. Advancements in smart devices have enabled the real-time collection of information about users’ physiological parameters [12]. Moreover, digital technologies offer remote exercise and personalised training programmes [13]. The Internet of Things (IoT) constitutes a network of interconnected devices designed to gather and exchange data, enhance the user experience and deliver personalised services. In the realm of tele-exercise and telemonitoring patients, IoT devices involve smartwatches, activity trackers, smart scales and environmental sensors [14]. Artificial Intelligence (AI) has the capability to analyse data gathered from wearable devices and environmental sensors, enabling it to offer personalised recommendations and feedback [15]. Finally, digitally live-streamed home exercises, such as virtual gym lessons, promote the development of customised programmes. To maintain good fitness levels, utilizing mobile fitness APPs for live-streamed tele-exercise, guided workout sessions with live instructors, personalised programmes, activity tracking, video conferencing, and remote health monitoring can be offered (multifactorial intervention) [16].

Different ages of participants could be involved in this process, from children and adolescents, who use different APPs to increase active time even in a school setting [17] and are enthusiastic about using technological devices [18], to elderly people, who tend to engage in sedentary behaviours and could benefit from using mobile fitness APPs [19], reducing barriers to PA and concerns about leaving homes [20]. No gender-based differences were observed in the frequency of utilising apps to monitor or promote PA [21]; however, the impact of increasing PA was more pronounced in females than in males [22].

The aim of this study was to identify the most accessible and effective strategy for promoting PA using APPs and other devices in healthy people or chronic disease patients.

It could be hypothesised that the use of APPs would primarily promote a change in lifestyle habits that could become a routine for a more active and healthier lifestyle.

## 2. Material and Methods

### Information Sources and Search Strategy

To identify relevant articles, the following online databases were used: Scopus; PubMed, EBSCOhost, and Web of Science (Figure 1).

In addition, the citations and reference lists of the retrieved relevant reviews were screened for any further relevant publications. The search was conducted in June 2023. The database was queried to search for the following terms in the articles’ titles, abstracts, and keywords (Table 1).

## 3. Results

### 3.1. Weight Loss and Counteracting Obesity through Tele-Exercise

The problem of overweight and obesity appears to be one of the most urgent problems in many western countries, with a profound impact on quality of life and implications for both physical and mental health [23], as well as significant impacts on worldwide healthcare economic expenditure [24]. It is also a risk factor in several pathologies [25,26,27].

The worldwide occurrence of obesity has nearly doubled in recent decades, emerging as a substantial public health issue [28]. In 2016, over 1.9 billion individuals aged 18 years and older (39% men and 40% women) were classified as overweight, with more than 650 million of them meeting the criteria for obesity. The prevalence of obesity almost tripled between 1975 and 2016. Obesity is acknowledged as a modifiable risk factor for cardiovascular diseases and overall mortality [29].

The risk of weight gain is mainly connected to an insufficient daily level of PA at all ages. APPs can concurrently monitor and address different factors associated with weight loss, such as tracking PA, monitoring dietary intake, offering counselling and delivering motivational messages, all within a multifaceted approach [30]. The use of APPs suggesting tailored activities to address individual needs has proven to be effective in attaining the objective of increasing daily PA and weight loss [12]. Additional APP functionalities, including short-term goal setting, walking programmes or home-based exercises, have demonstrated weight reduction at 12 weeks [31]. Balk-Moller et al. [32] showed that the psychological and motivational aspects of the use of devices, such as APPs, had a positive impact on weight loss in a 16-week training programme. The platform includes social support features such as rewards, team competitions, push notifications and a forum facilitating interactions among multiple users, thereby enhancing motivation for participation and APP utilisation.

Furthermore, when comparing the effects of home-based PA, following APP indications, and supervised PA, performed in a face-to-face setting, Berglind et al. [31] found that both options were equally effective. Moreover, both long-term interventions, focused on increasing daily step counts [33] or organising online face-to-face lessons, have been found to be effective in sustaining user motivation [31].

Multifactorial interventions, such as combining smartphone technology with text messaging [34,35] or APP counselling [36], have proven to be more effective than single interventions.

Several interventions based on multifunctional APPs have been developed to increase adherence to an active, healthy lifestyle, despite a limited or moderate effect on body weight reduction [37,38].

The levels of PA showed a significant enhancement through the utilisation of platforms, which offer opportunities for interaction among individuals. Creating communities on websites and social media, with people with a common goal, sending emails and text messages, seems to be successful in enhancing compliance with a complete weight loss treatment. The participants experienced improvements in weight loss and body fat reduction, as well as enhancements in systolic and diastolic blood pressure and improvements in the lipid profile [37]. Participants, when using platforms, have the option to select various customised workout routines tailored to their specific needs, personal fitness parameters and goals. Social interaction among multiple users, including team competitions, as well as interactions between a user and a coach with access to personalised programmes, have demonstrated substantial improvements in terms of increased daily PA and weight reduction within a 9-month timeframe [32].

The interaction between a technological device and individual lifestyle habit monitoring, using Fitbit or smart bands, achieved good effects on health and behavioural changes [39,40].

In conclusion, the usefulness of APPs in promoting increased daily activity and weight loss has been confirmed; however, it is essential to carefully assess what approach is most suitable for different individual needs [41]. It is crucial to acknowledge that the effectiveness of APPs can vary depending on the specific characteristics of the APP, the support offered by industry professionals, the stimulation of interactions and a sense of competition among participants. Moreover, these findings underscore the significance of combining PA monitoring with sufficient motivational support to optimise weight loss and encourage sustained active lifestyles in the long term [42].

Probably the most significant positive goal in the use of APPs lies in being a determining factor in changing lifestyle habits and consciously promoting a healthy lifestyle even after an individual has stopped using them [43].

### 3.2. Advanced Strategies for Cardiovascular Disease Control

The significant advancement in technology in the area of home telemonitoring and tele-exercise has enabled a substantial improvement in cardiovascular disease (CVD) management.

CVDs are considered one of the main causes of death and disability worldwide, with an estimated 64.3 million people living with heart failure, and correlated risk factors such as hypertension, high cholesterol and atherosclerosis could lead to heart failure, arrhythmia and stroke [26].

Prolonged and regular adherence to PA plays a fundamental role in the prevention and reduction of cardiovascular risk factors [44], and an active lifestyle is an effective strategy for reducing the consumption of antihypertensive drugs [45,46]. Nevertheless, in the majority of cases, behavioural changes aimed at adopting a healthy lifestyle are frequently discontinued over a long time, leading to a reduction in the levels of PA that have been attained.

Several studies [47,48,49] have shown the positive impact of mobile health APPs (mHealth) on the health and daily behavioural changes of individuals with CVD, in particular, through the promotion of and an increase in PA levels [50,51]. The American Heart Association has formally acknowledged the legitimacy of mHealth, defining it as the use of digital technologies for the provision of health services and information [50]. Therefore, the use of APPs can be considered an excellent tool for CVD prevention [51].

Several studies on this topic have yielded different results. Lai et al. [52] did not find significant differences in hypertension control between older adults who used digital technology and those who did not. Conversely, Dunlay [53] demonstrated that a programme combining standard cardiac rehabilitation with a mobile-device-based rehabilitation programme had a positive impact on risk factor profiles, lifestyle habits and reduced hospitalisations in patients with acute coronary syndrome. Johnston [54] found that patients with myocardial infarction participating in smartphone-based interactive treatment showed greater adherence to the intervention and a reduction in drug intake, with a trend towards improving cardiovascular outcomes and quality of life.

Additional support for the use of smartphone APPs in cardiac rehabilitation is provided by the meta-analysis conducted by Zhou [55,56], highlighting that smartphone-assisted cardiac rehabilitation significantly enhanced the peak levels of oxygen consumption achieved with graded exercise to exhaustion (VO2peak) in patients with cardiovascular diseases, compared to conventional cardiac rehabilitation or usual care.

A multifactorial intervention in self-monitoring combined with setting up counselling, which was implemented via the internet, resulted in favourable outcomes for cardiovascular health [57]. Interactive PA based on a new home intervention system with specialised software and tablet computers wirelessly connected to each other may substantially improve accessibility and increase knowledge around cardiovascular problems, reducing hospitalisation [58]. The use of AI-driven interactions, in combination with monitoring, has been applied in different contexts. Persell et al. [59] employed a smartphone APP featuring conversational AI that incorporated cognitive-behavioural therapy techniques. This APP was used to deliver support and coaching for the self-management of hypertension and the promotion of healthy behaviour, including dietary choices, PA, medication adherence, blood pressure monitoring, sleep management and stress management. Users were allowed to set PA goals and received coaching on progress towards these goals.

In conclusion, telerehabilitation is confirmed as an efficient strategy to reduce both CVD and mortality. APPs for electronic communication and information technologies within the healthcare domain (e-Health), in terms of prevention and long-term CVD management, are effective in containing cardiovascular risk factors and reducing the occurrence of cardiac events [60]. The benefits of mobile health-based cardiac rehabilitation programmes persist both during the rehabilitation period and in the subsequent three months. These improvements are more favoured by personalised approaches, considering patients’ age and gender and the level of disease that counteracts the possibility of rehospitalisation [61].

Moreover, it could be taken into consideration that the intervention cost of tele-exercise in each modality is relatively low [62].

### 3.3. Diabetes Self-Management through Mobile APP-Based Interventions

Diabetes is a multifaceted chronic disease that affects millions of individuals worldwide. Policymakers and healthcare providers assert that only direct patient self-management can counteract the progression of the disease and, consequently, improve patient-related outcomes [63]. On a global scale, healthcare expenditure related to diabetes was estimated at USD 760 billion in 2019, and it is expected to grow to USD 825 billion annually by 2030 [64].

Mobile APPs represent a widely available and low-cost technology and can be used to engage diabetic individuals in lifestyle changes, facilitating health control. Regarding cost-effectiveness evaluations, mobile APPs showed a net cost savings of 8.8 percent with respect to a traditional approach [65], confirming their economic viability [33].

A high level of daily PA, along with effective stress management, enhances emotional well-being and promotes weight loss, which is an important preventive factor associated with positive diabetes outcomes, as the American Diabetes Association guidelines highlight [66].

The support that APPs can provide in managing diabetes involves two aspects: on the one hand, these devices aid in self-monitoring health parameters such as blood glucose levels and Glycated Haemoglobin (HbA1c) [67], and on the other hand, APPs may promote a lifestyle that helps to protect against the progression of the disease, soliciting healthy eating, a good amount of PA and weight management [68].

Several studies have confirmed the benefits of using mobile APPs in diabetes management. Fukuoka et al. [33] found enhancements in diabetes risk factors, such as weight loss and a reduction in blood pressure, after five months of PA using a mobile APP, which provided daily reminders, and a pedometer. Significant improvements were also achieved using the mDiabetes APP system [69]. Conversely, Jospe [25], after studying the self-monitoring of weight and diet using APPs, did not find weight loss or changes in dietary habits; nevertheless, an improvement in the patients’ depression and anxiety was observed, showing an enhancement in the psychosocial care of the patients [68].

Moreover, APP-interventions that combine glucose monitoring, diet, PA and a clinical decision support system have proven to be more effective in improving both health education and diabetes outcomes than single interventions [69,70]. Alonso-Dominguez [71] proposed a multifactorial intervention for diabetic patients combining smartphone APPs that suggested guidelines for PA and healthy diet and the use of pedometers. The increase in Metabolic Equivalent of Tasks (METs min/week) and the adherence to the programme confirmed the validity of this proposal. High-frequency and interactive communication between patients and providers resulted in greater reductions in HbA1c compared to approaches with unidirectional data communication [72]. The telerehabilitation programme for patients with type 2 diabetes mellitus was proven to be safe and effective in improving glucose control, physical fitness and psychosocial status [73].

Despite variations in study methodologies, it can be concluded that multifactorial APP interventions [69] are the most effective method for enhancing health education and diabetes outcomes [70]. APPs that include counselling on a regular daily dose of PA, a balanced diet, and practices related to sleep and leisure are commonly linked to Behaviour Change Techniques (BCTs) [43]. User engagement in goal setting, the self-monitoring of behaviours, feedback on behaviours, prompts/cues, rewards and social support could promote a better long-term lifestyle [74].

### 3.4. Promoting Regular Physical Activity via APPs for Cancer Patients

In 2020, 19.292.789 new cancer cases were reported worldwide, and 9.958.133 deaths were recorded. Among the most common cancer types, breast cancer and lung cancer rank first and second, with a percentage of cases at 11.7% and 11.4%, respectively. Other cancer types with significant incidence rates include colorectal cancer (10%), prostate cancer (7.3%), and stomach cancer (5.6%) [29].

Cancer cases are increasing worldwide despite improvements in treatment reducing the number of fatalities. Implementing preventive measures has become mandatory, recommending an active and healthy lifestyle to enhance the efficacy of therapeutic interventions and the tolerability of side effects such as fatigue [75].

PA is a useful complementary non-pharmacological approach in the prevention and treatment of different types of cancer [76]. Prolonged periods of inactivity increase the risk of cancer and worsen chronic conditions, while regular PA mitigates the onset of specific cancer types [77]. Consequently, sustaining PA is of vital importance for cancer survivors who have completed their primary treatment [78] and might contribute to increased survival rates and decreased chances of recurrence [79]. Recently, cancer survivors have been recommended to adhere to the same PA guidelines as the general adult population, that include to be engaged at least in 150 min of moderate-intensity PA and two sessions of resistance-based training per week [80]. The mechanisms by which PA reduces the risk of cancer are not yet understood. The effects of PA on carcinogenesis are likely to be multifactorial and influenced by individual characteristics such as age and gender, as well as specific exercise-related factors, including the type, intensity and frequency of PA [81]. The use of technology presents a cost-effective opportunity to facilitate more active lifestyles for cancer survivors [82]. Digital interventions have demonstrated promise in encouraging PA among individuals who have survived cancer, although the majority of the interventions have been web-based, and only a limited number of studies have incorporated mobile APPs [83].

There is general consensus that an APP focused on walking would have the highest appeal for cancer survivors [84]. Walking is considered safe, accessible and attainable for the majority of patients, regardless of their physical condition, cancer subtype, treatment regimen or associated side effects [83]. There was a prevailing belief that resistance-based training using APPs was excessively challenging and potentially unsafe, especially at the introductory levels of APP use [83]. Conversely, a telemedicine approach involving inspiratory muscle training and walking has proven to be acceptable and safe. This approach has demonstrated the potential to break the vicious cycle of “dyspnea-inactivity” among lung cancer survivors [85].

Finally, the intervention, based on an online community, allowed participants to view the daily step counts of other members, providing mutual motivation and promoting health-related activities [86]. Health-related information regarding diet and PA could be posted on the community board, improving motivation and mental distress [87]. mhealth, based on a self-help group, may enhance autonomy in organising each PA intervention tailored to specific individual needs and promote the sharing of activities, needs and objectives [88].

It could be concluded that the use of APPs and other technological support allows for enhanced PA levels and active lifestyles among cancer survivors, influencing their engagement through digital interventions. A counselling service could help patients choose Appropriate Apple (iOS) and Android mHealth APPs based on the type of PA preferred, behavioural suggested strategies and other characteristics and preferences [89].

### 3.5. Self-Management of Osteoporosis: Telemedicine as an Optimal Strategy

Osteoporosis is a non-communicable and prevalent bone disorder, impacting approximately one in three women and one in five men aged 50 and older worldwide [90].

It has been estimated that 151 osteoporosis cases cause up to 9 million fractures annually worldwide [91]. Primary osteoporosis is associated with age and hormonal imbalances related to gender. In the five years following menopause, women can lose over 30% of their bone mass. Secondary osteoporosis is induced by various concurrent medical conditions and/or pharmaceutical treatments [92]. The primary goal of osteoporosis management should be focused on prevention. The loss of bone structure occurs progressively and often without specific symptoms and frequently leads to a lack of initiated therapies or prevention strategies [93]. A behavioural modification strategy that has shown effectiveness in addressing osteoporosis is self-management. Self-management, particularly when facilitated through smartphone APPs for osteoporosis prevention, typically involves activities such as goal setting, self-monitoring and adopting specific behaviours [94]. In addition to PA, maintaining a healthy diet, adhering to medication treatment and preventing falls are included in this approach, which is linked to changes in health behaviour [95]. The PA methodology, particularly involving high-impact activities and weight bearing, is crucial for optimising the accumulation of bone mass [96]. Previous studies that have employed PA to prevent osteoporosis through wearable technology and/or a mHealth APP have found good results in enhancing bone mineral density, despite consisting of unsupervised PA interventions [97,98]. Moreover, smartphone APPs could be used to easily detect a history of falls through dynamic real-time measurements [99]. However, it is crucial to note that many of the available APPs lack clinical validation, which may compromise their effectiveness in detecting results and the validity of the indications [100].

Small but positive outcomes have been identified in interventions based on PA implementation through mobile APPs in the management and mitigation of osteoporosis [101]. Despite the lack of statistically significant outcomes in several studies, it was observed that active individuals using APPs experienced a lower decrease in bone density compared to the national average for women and men in the corresponding age group [98,102]. It is possible that not all participants consistently followed the indications provided by the mobile APPs, or they may deviate from the intended use of the interventions outlined in the instructions. Crucial to achieving positive objectives, even in the absence of active participation, is sustained retention, following the recommendations on frequency, intensity, duration and typology of activity and exercises precisely [101].

Additionally, the use of APPs solicited changes in general health behaviour and social facilitation, which is likely to stimulate internally driven motivation and enhances a sense of ownership and participation, consequently increasing overall effectiveness [82]. In conclusion, the use of APPs results in better management of osteoporosis when the application allows contact with the user, either through reminders to engage in specific activities or by sending encouraging messages. This integration combines the knowledge and expertise of healthcare professionals with the personal preferences of individuals [103].

### 3.6. App for Cognitive Training in Dementia

The ageing of the global population has led to an increase in the incidence of chronic diseases, including dementia. The prevalence of this condition is on the rise, and in 2020, more than 55 million people were living with dementia worldwide, with the prospect of doubling every 20 years [104]. In response to this growing challenge, early diagnosis and preventive measures have become increasingly crucial, as highlighted by recent epidemiological research [105].

Several systematic reviews and meta-analyses have indicated that both cognitive training and physical exercise can slow cognitive decline, showing positive effects ranging from mild to moderate on global cognitive function. Higher levels of PA are associated with benefits such as superior cognitive function, a larger hippocampal volume and higher levels of neurobiological health markers compared to those leading a sedentary lifestyle [106,107].

Additional promise arises from the use of APPs as a tool to address cognitive decline. There are few studies that have employed the utilisation of mobile APPs to encourage PA in individuals predisposed to cognitive decline [108]. Considering that the strategy of promoting PA through APPs for this population allows for a combination of cognitive and physical exercises, enhancing effectiveness and adherence to training programmes [109,110], further studies are needed to thoroughly explore and substantiate the clinical evidence.

## 4. Conclusions

The use of APPs and technologies offers significant opportunities for improving people’s health and well-being through daily PA enhancement and chronic disease management, given features such as accessibility and convenience. People, by interacting with these APPs, can monitor their activities wherever they are, performing PA even if they cannot easily access gyms or specific sports facilities. Real-time monitoring of PA and physiological metrics provides users with immediate feedback on their performance, enabling them to become proficient in making corrections.

The personalisation of protocols, especially in the management of chronic diseases such as cancer, diabetes and osteoporosis, can enhance the effectiveness of training. Furthermore, continuous motivational support with goal setting, challenges and social interaction significantly reduces the dropout rate. Moreover, creating an online group workout could have a positive impact by fostering healthy competition that motivates individuals to achieve their goals [111]. Nevertheless, this goal can be more easily achieved by using advanced technological equipment, such as platforms connecting patients with each other, with expert coaches or with AI.

Especially in chronic disease management with APPs, the multifactorial interventions (apps + counselling) achieved more relevant improvements in PA levels and mental well-being [112]. Moreover, as has emerged in the recent literature, we can emphasise that the extended use of health APPs and telehealth may cause lasting changes in an individual’s mindset towards a healthier lifestyle, even after they stop using these devices. Different behaviour change techniques are integrated into mobile health APPs, correlating with enhanced user engagement.

Despite these promising results, several aspects that emerged from these findings should be evaluated. Firstly, should be considered that the lack of supervision by experienced professionals in PA administration, and the self-managed activity of a patient may not be sustainable over time, reducing the effects. Secondly, the present review highlighted that most of the studies derived their results from short-term interventions. This aspect should not be underestimated; for instance, concerns like overuse injuries, known as “sur ménage,” may not emerge from short-term exercise interventions [113], and short-term interventions, in the phase of transitioning from sedentary to active lifestyles, yield rapid initial benefits; however, sustained fitness improvement requires personalised training over time [114].

Another important issue concerns the relationship between humans and devices. The frequent use of smartphones and other devices may lead to compulsive behaviours related to PA, diet and constant checking due to dependency on notifications. Evidence suggests that people can develop smartphone dependency and habitual checking behaviours, contributing to a phenomenon known as “Smartphone Addiction” [115,116,117].

Some limitations have to be addressed:
Generalising the results may overlook individual and population differences.The quality of the APPs is not thoroughly discussed, and attention to the long-term sustainability of interventions is limited. User adherence to APP recommendations and the potential for alteration are acknowledged.


## Figures and Tables

**Figure 1 ijerph-21-00233-f001:**
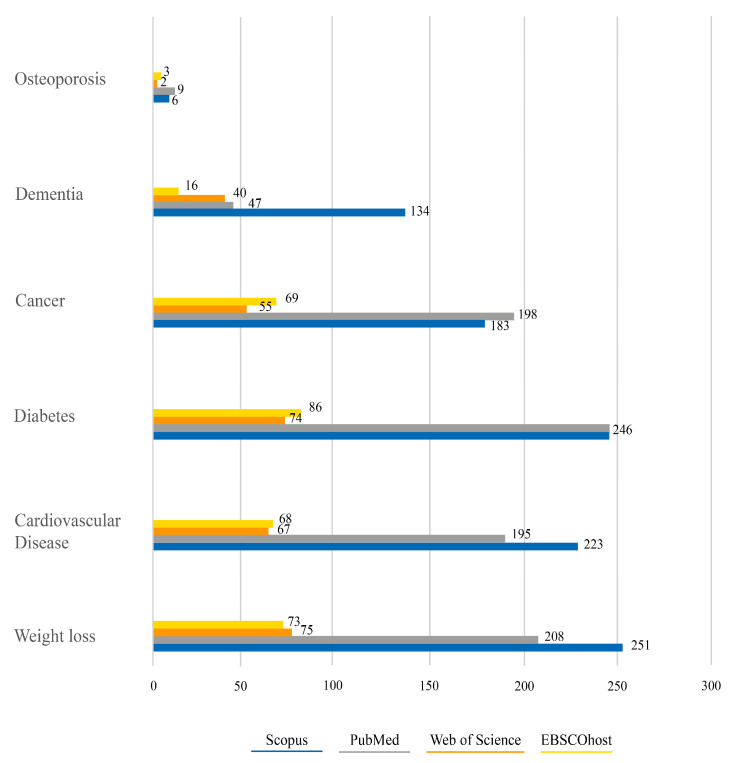
Detection of articles on the “mHealth” application about PA in managing osteoporosis, dementia, cancer, diabetes, cardiovascular diseases, and weight loss over the last 10 years via Scopus, PubMed, Web of Science, and EBSCOhost databases.

**Table 1 ijerph-21-00233-t001:** Full search strategy for database.

Database	Specificities of the Database	Search Strategy
Scopus PubMed EBSCO Web of Science	Search for title and abstract also includes keywords	(“mHealth” OR “Mobile Health” OR “E-health”) AND (“App” OR “Smartphone application”) AND (“Mobile” OR “Smartphone”) AND (“Physical Activity” OR “Exercise”) AND (“Health Education” OR “Patient Education”) AND (“Self-monitoring” OR “Self-tracking”) AND (“Steps” OR “Step count”) AND (“Disease” OR “Illness” OR “Condition”) AND (“Weight loss” OR “Weight management”) AND (“Cardiovascular disease” OR “Heart disease”) AND (“Diabetes” OR “Diabetic”) AND (“Cancer” OR “Oncology”) AND (“Osteoporosis” OR “Bone health”) AND (“Cognitive decline” OR “Cognitive function”) NOT (“Smartphone” AND “Physical Activity”)

## Data Availability

Data are available upon request to the corresponding author.

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
