# Peer review of "Enhancing Behavioural Changes: A Narrative Review on the Effectiveness of a Multifactorial APP-Based Intervention Integrating Physical Activity"

_ijerph, 2024, doi:10.3390/ijerph21020233_

Round 1

Reviewer 1 Report

Comments and Suggestions for Authors

Line                 Comment

Page 4 end of section 3: What is “LEON” ?

Beginning of Page 5: There is an extra space and “technologies” alone seems too broad, AMA specifies Mobile and Communication tech.

Page 5 first full paragraph, first sentence: change “has” to “have.” Also, last sentence change “great” to greater.”

Page 5 second paragraph: Define VO2peak. E.g. peak levels of oxygen consumption achieved with graded exercise to exhaustion.

Page 5 end of section 4: define e-Health.

Page 6 define METs

Page 7 first sentence of first full paragraph: “Cheung” not needed.

Page 7 second paragraph: change M-health to mHealth ?

Page 7 last sentence of section 6: extra space.

Page 8 last sentence of section 8: Not sure what this sentence is in reference to.

Page 9 middle: what does BHC stand for?

Author Response

Page 4 end of section 3: What is “LEON” ?

Deleted. It's a mistake. (Page 5)

Beginning of Page 5: There is an extra space and “technologies” alone seems too broad, AMA specifies Mobile and Communication tech.

The extra space has been removed. We have replaced technologies with digital health technologies as specified by the AMA

Page 5 first full paragraph, first sentence: change “has” to “have.” Also, last sentence change “great” to greater.”

Done.

Page 5 second paragraph: Define VO2peak. E.g. peak levels of oxygen consumption achieved with graded exercise to exhaustion.

Done

Page 5 end of section 4: define e-Health.

Done

Page 6 define METs

Done

Page 7 first sentence of first full paragraph: “Cheung” not needed.

Deleted.

Page 7 second paragraph: change M-health to mHealth ?

The correction has been made. The words have the same meaning; a capital letter was used exclusively because the word 'm-health' followed a period.

Page 7 last sentence of section 6: extra space.

The extra space has been removed

Page 8 last sentence of section 8: Not sure what this sentence is in reference to.

The sentences has been deleted

Page 9 middle: what does BHC stand for?

The sentence has been rephrased.  

Reviewer 2 Report

Comments and Suggestions for Authors

Dear authors,

The research paper analyzes the impact and effectiveness of mobile applications in promoting an active lifestyle and managing chronic diseases. Additionally, the paper highlights the advantages of technology, as well as its critical aspects such as device dependency and ethical issues. I believe this is a relevant topic in the current context, where technology plays an increasingly important role in healthcare.

However, I consider that the paper has some shortcomings, which I would like to mention:

1.    I believe that the term "physical activity" should be included in the title, as a significant part of this study is based on organizing and managing physical activity through various applications.

2.    In the keywords section, please specify that it is related to physical exercise, as exercise can take various forms such as mathematical, writing, reading, etc.

3. In the Introduction chapter, when mentioning ".......... involving exercise interventions," what type of exercise are you referring to? Physical exercise? If yes, please specify this aspect.

4.    According to the journal's recommendations, "all manuscripts must contain the required sections: Author Information, Abstract, Keywords, Introduction, Materials & Methods, Results, Conclusions, Figures and Tables with Captions, Funding Information, Author Contributions, Conflict of Interest and other Ethics Statements," available at https://www.mdpi.com/journal/ijerph/instructions#preparation. I consider this structure should be adhered to.

5. In the Information sources and search strategy section, you mentioned four online databases, but you only presented the findings for Scopus. Why didn't you provide information for the other three databases? I think it would be relevant to know how many publications were found on this topic and what their dynamics were.

6.   In the Weight loss and counteracting obesity through tele-exercise section, you mentioned, "Balk-Moller et al. [30] showed that the psychological and motivational aspects...." I believe it would be useful to provide examples of some of these psychological and motivational aspects of the applications in the given context. Additionally, in this section, statements like "The levels of PA showed a significant enhancement...., .... seems to be successful in enhancing compliance......, .... have demonstrated substantial improvements....." should be supported with specific numerical data or percentages from the cited sources to convince the reader.

7.   All presented sections should be supplemented with numerical data when making certain claims (e.g., in the Diabetes Self-Management through mobile APP-based interventions section, I believe you did the right thing when you made the statement "On a global scale, healthcare expenditure related to diabetes was estimated at 760 billion dollars in 2019, and it is expected to grow to 825 billion dollars annually by 2030 [62].").

8.   Moreover, I think data about each investigated section from all four studied platforms should be extracted, providing a quantitative measure per platform regarding how many studies address each respective section. For instance, how many studies were found for Weight loss and counteracting obesity through tele-exercise in Scopus, Web of Science, Pub Med, and EBSCO? How many for Advanced strategies for cardiovascular disease control, how many for Diabetes Self-Management through mobile APP-based interventions, etc.?

Best regards!

Author Response

  1. I believe that the term "physical activity" should be included in the title, as a significant part of this study is based on organizing and managing physical activity through various applications.

The title has been modified and integrated with the term 'Physical Activity' as you suggested

  1. In the keywords section, please specify that it is related to physical exercise, as exercise can take various forms such as mathematical, writing, reading, etc.

Done.

  1. In the Introduction chapter, when mentioning ".......... involving exercise interventions," what type of exercise are you referring to? Physical exercise? If yes, please specify this aspect.

Done.

  1. According to the journal's recommendations, "all manuscripts must contain the required sections: Author Information, Abstract, Keywords, Introduction, Materials & Methods, Results, Conclusions, Figures and Tables with Captions, Funding Information, Author Contributions, Conflict of Interest and other Ethics Statements," available at https://www.mdpi.com/journal/ijerph/instructions#preparation. I consider this structure should be adhered to.

The manuscript formatting has been modified according to the journal’s recommendations.

5. In the Information sources and search strategy section, you mentioned four online databases, but you only presented the findings for Scopus. Why didn't you provide information for the other three databases? I think it would be relevant to know how many publications were found on this topic and what their dynamics were.

The chart on Scopus was included solely as an example, so we deleted it to avoid misunderstandings concerning the other databases.

6. In the Weight loss and counteracting obesity through tele-exercise section, you mentioned, "Balk-Moller et al. [30] showed that the psychological and motivational aspects...." I believe it would be useful to provide examples of some of these psychological and motivational aspects of the applications in the given context. Additionally, in this section, statements like "The levels of PA showed a significant enhancement...., .... seems to be successful in enhancing compliance......, .... have demonstrated substantial improvements....." should be supported with specific numerical data or percentages from the cited sources to convince the reader.

Done. The psychological and motivational aspects have been added as you suggested.

7. All presented sections should be supplemented with numerical data when making certain claims (e.g., in the Diabetes Self-Management through mobile APP-based interventions section, I believe you did the right thing when you made the statement "On a global scale, healthcare expenditure related to diabetes was estimated at 760 billion dollars in 2019, and it is expected to grow to 825 billion dollars annually by 2030 [62].").

The sections have been integrated as you suggested.

8. Moreover, I think data about each investigated section from all four studied platforms should be extracted, providing a quantitative measure per platform regarding how many studies address each respective section. For instance, how many studies were found for Weight loss and counteracting obesity through tele-exercise in Scopus, Web of Science, Pub Med, and EBSCO? How many for Advanced strategies for cardiovascular disease control, how many for Diabetes Self-Management through mobile APP-based interventions, etc.?

Considering that this is not a systematic review, we have not reported these data. The number of publications and the number of databases were merely used to gather information and materials on the topic, neverthless the databases were not systematically searched.

Reviewer 3 Report

Comments and Suggestions for Authors

Dear authors,

the manuscript do not have any novelty about this topic , although the manuscript is well written and structured. You aimed "to identify the most accessible and effective strategy in promoting PA using APPs and other devices in healthy people or chronic disease patients." and performed an search in some of the most important dabases. I would had EMBASE, sportdiscuss and cochrane. In conclusion i suggest to remove all the references and rewrite this section with only your fundamental conclusions. 

Also, in limitations you refer : "Ethical considerations regarding privacy and data security deserve further attention." this statmment should be removed since all articles should have an ethics committie aproval befora aplication that protect all this issues. 

Congratulations

Author Response

the manuscript do not have any novelty about this topic , although the manuscript is well written and structured. You aimed "to identify the most accessible and effective strategy in promoting PA using APPs and other devices in healthy people or chronic disease patients." and performed an search in some of the most important dabases. I would had EMBASE, sportdiscuss and cochrane. In conclusion i suggest to remove all the references and rewrite this section with only your fundamental conclusions. 

In our opinion, two innovative aspects emerged from this narrative review: the first is the greater effectiveness of multifactorial interventions delivered through mHealth, the second consists of lifestyle changes induced by the use of apps, which persist even after the interruption of their use. We have tried to highlight this in the conclusions paragraph, as you suggested.

Also, in limitations you refer : "Ethical considerations regarding privacy and data security deserve further attention." this statmment should be removed since all articles should have an ethics committie aproval befora aplication that protect all this issues. 

Following your suggestion, the statement has been deleted.

Reviewer 4 Report

Comments and Suggestions for Authors

The topic addressed in this manuscript may interest the IJERPH readers. The study provides a summary of intervention studies using telehealth and APPS technologies in promoting behavioral changes in various chronic disease populations. This type of review is needed to allow for a better understanding of the technology.

Major concerns

Although the review is divided into several sections based on chronic diseases, obesity, cardiovascular disease, diabetes, cancer, osteoporosis, and dementia, the idea is the same throughout each section with different wording, with the use of APPS could lead to an increase in physical activity and lead to better health outcomes. This becomes repetitive, leading to an inconsistency in the title regarding behavior change. In addition, in the obesity section, the content is less about the intervention and more about the function of wearable devices and APPS in promoting physical activity. Authors should consider a separate section focusing on the features of wearable devices and APPS. In my understanding, pedometers might not fit into the realm of APPS or telehealth because many pedometers do not involve the use of telecommunication devices. I recommended the authors provide more in-depth details of each intervention and how the telehealth devices were used in the interventions. In each of the sections. The authors mentioned that telehealth devices were used in the intervention, however, it did not provide any in-depth information on how the devices were used. This study is also very similar to many previously published articles on the use of wearable devices to promote physical activity. If the authors could provide more in-depth details on the methodology of using telehealth devices, then I think it could provide much more practical strategies for using the devices.

Author Response

Although the review is divided into several sections based on chronic diseases, obesity, cardiovascular disease, diabetes, cancer, osteoporosis, and dementia, the idea is the same throughout each section with different wording, with the use of APPS could lead to an increase in physical activity and lead to better health outcomes. This becomes repetitive, leading to an inconsistency in the title regarding behavior change. In addition, in the obesity section, the content is less about the intervention and more about the function of wearable devices and APPS in promoting physical activity. Authors should consider a separate section focusing on the features of wearable devices and APPS. In my understanding, pedometers might not fit into the realm of APPS or telehealth because many pedometers do not involve the use of telecommunication devices. I recommended the authors provide more in-depth details of each intervention and how the telehealth devices were used in the interventions. In each of the sections. The authors mentioned that telehealth devices were used in the intervention, however, it did not provide any in-depth information on how the devices were used. This study is also very similar to many previously published articles on the use of wearable devices to promote physical activity. If the authors could provide more in-depth details on the methodology of using telehealth devices, then I think it could provide much more practical strategies for using the devices.

Pedometers represent the simplest technological devices used to encourage physical activity. In fact, many studies have employed them in combination with other interventions. However, following your advice, we have attempted to exclude studies that utilized this instrumentation.

The examination of the majority of articles utilizing physical activity prompted by technological devices to counteract chronic diseases revealed a behavior change among participants related to adopting healthier and more active lifestyles: this was the primary outcome of the review, as reported in the title. The second consideration emerging from the review, more methodological in nature, demonstrated that complex and multifactorial interventions, combining counseling, nutritional advice, and physical activity, proved to be more effective. Reporting these results in different paragraphs may seem somewhat repetitive but is necessary to substantiate the conclusions

Round 2

Reviewer 2 Report

Comments and Suggestions for Authors

Dear authors,

Thank you for considering my suggestions, and I believe the paper has been slightly improved. However, I would like to address the following shortcomings:

  • In the section "Information sources and search strategy," I believe that the data about the sources studied from the four databases are very relevant since the paper is a narrative review, and they should not have been removed. It is important to know the number of sources studied and their dynamics over the last decade (the past 10 years), as you presented in the first version of the study.
  • Additionally, I think that data should be included for each investigated section from all four platforms studied, namely providing numerical data/platform on how many studies address the respective section. For example, how many papers were there for "Weight loss and counteracting obesity through tele-exercise" in Scopus, Web of Science, PubMed, and EBSCO, how many for "Advanced strategies for cardiovascular disease control," how many for "Diabetes Self-Management through mobile APP-based interventions," etc.

   Best regards!

Author Response

  • In the section "Information sources and search strategy," I believe that the data about the sources studied from the four databases are very relevant since the paper is a narrative review, and they should not have been removed. It is important to know the number of sources studied and their dynamics over the last decade (the past 10 years), as you presented in the first version of the study.
  • Additionally, I think that data should be included for each investigated section from all four platforms studied, namely providing numerical data/platform on how many studies address the respective section. For example, how many papers were there for "Weight loss and counteracting obesity through tele-exercise" in Scopus, Web of Science, PubMed, and EBSCO, how many for "Advanced strategies for cardiovascular disease control," how many for "Diabetes Self-Management through mobile APP-based interventions," etc.

In the "Information sources and search strategy" section, we have integrated the requested information by structuring a table that includes details related to the examined areas in reference to the four search platforms.